# Antifungal Susceptibility Data and Epidemiological Distribution of *Candida* spp.: An In Vitro Five-Year Evaluation at University Hospital Policlinico of Catania and a Comprehensive Literature Review

**DOI:** 10.3390/antibiotics13100914

**Published:** 2024-09-24

**Authors:** Maddalena Calvo, Guido Scalia, Laura Trovato

**Affiliations:** 1Department of Biomedical and Biotechnological Sciences, University of Catania, 95123 Catania, Italy; maddalenacalvo@gmail.com (M.C.); lido@unict.it (G.S.); 2U.O.C. Laboratory Analysis Unit, A.O.U. Policlinico “G. Rodolico-San Marco” Catania, 95123 Catania, Italy

**Keywords:** *Candida* spp., antifungal resistance, antifungal susceptibility testing, candidaemia

## Abstract

Background: Invasive fungal infections represent a concerning healthcare issue, with *Candida* spp. reported as the main aetiological agent. *Candida* spp. bloodstream infections show high mortality rates, indicating increasing antifungal-resistance episodes as a contributing feature. Despite the global prevalence of *C. albicans*, non-*albicans* species emerged as significant in the last decades. Methods: The present manuscript reports a five-year evaluation on *Candida* spp. bloodstream isolates and their antifungal susceptibility profiles, aiming to enrich the literature and epidemiological data. Results: According to the gathered data, antifungal-resistance cases remained uncommon. However, the study revealed rare resistance phenotypes such as a single case of pan-echinocandin resistance *C. albicans*. Conclusions: Finally, a comprehensive review of *Candida* spp. antifungal resistance integrates the data, emphasizing the extreme species-specific variability and the consequent importance of always providing species identification.

## 1. Introduction

Invasive fungal infections represent a significant healthcare challenge, especially regarding immunocompromised and intensive care patients. Fungal bloodstream infections mainly report *Candida* spp. as the aetiological agent, documenting a significant prevalence in the United States and Europe [1]. *Candida* spp. bloodstream infections often reach elevated mortality rates due to fungal virulence and/or patients’ underlying conditions. For instance, critical patients suffer from risk factors such as prolonged hospitalization, neutropenia, and invasive surgical procedures [1,2]. Furthermore, *Candida* spp. infections may meet therapeutic failure depending on the eventual biofilm formation and antifungal-resistance mechanisms [1,3]. *Candida albicans* is the most isolated fungal species during candidaemia episodes and is known for its relevant capability to form biofilm. Despite this species’ global prevalence, recent epidemiological data document a continuous evolution to non-*albicans Candida* species. *Candida glabrata*, *Candida parapsilosis*, *Candida tropicalis*, and *Candida krusei* are responsible for numerous candidaemia cases. Additionally, uncommon species such as *Candida guilliermondii*, *Candida lusitaniae*, and *Candida kefyr* rarely cause systemic infections [1,4]. As regarding antifungal-resistance rates, all the above-mentioned *Candida* species may variably show resistance episodes through drug target alteration or reduced cellular drug concentration [4]. Antifungal-resistance mechanisms pose a significant therapeutical issue due to systemic infections’ severity and critical patients’ conditions [5]. Diagnostic workflows currently face low sensitivity rates and prolonged incubation intervals in the case of a bloodstream-infection microbiological diagnosis. Consequently, clinicians could proceed with empiric antifungal treatment before the specific options are established. These therapeutical protocols can lead to unnecessary antifungal usage and emerging resistance promotion [6].

The present manuscript reports a five-year evaluation on *Candida* spp. bloodstream isolates and their antifungal susceptibility profiles, aiming to enrich the literature and epidemiological data. A comprehensive review of *Candida* spp. resistance profiles completes the study, highlighting the importance of always integrating microbiological reports with species identification.

### 1.1. Antifungal Resistance in Candida albicans

Triazoles represent the major antifungal drug class in clinical usage. The main azole-resistance mechanism is efflux pumps overexpression, which leads to insufficient drug concentration within the fungal cell. Furthermore, ERG11 gene mutations contribute to altering the corresponding enzyme (lanosterol 14-α-Demethylase), avoiding the drug binding to its main target (ERG11) in the fungal cell structure [7,8]. *C. albicans* azole resistance demonstrated low worldwide rates, accounting for rare and limited outbreaks in North America and South America [7,8].

5-fluocytosin-resistance episodes are related to the uracil phosphoribosyltransferase enzyme, whose mutations definitively alter the 5-fluocytosin antifungal action. This resistance mechanism easily appears after drug exposition; thus, monotherapy regimens are not recommended, leading to several past changes in antifungal therapeutical plans [7,8].

Finally, mutations in the ERG3 gene lead to a significant reduction in the ergosterol concentration within the fungal cellular membrane. As a consequence, amphotericin B fails in its antifungal activity, facing difficulties in finding the main target. An increased catalase activity, along with a reduced susceptibility to the oxidative damage, may also contribute to the same resistance. However, amphotericin B-resistance mechanisms are uncommon, and this is an interesting advantage due to the increasing clinical importance of this molecule within antifungal therapeutical regimens [8].

Echinocandins (caspofungin, micafungin, and anidulafungin) are the first-line therapeutical choice in the case of *Candida* spp. systemic infections, due to their fungicidal activity, effectiveness, and safety. Echinocandins inhibit the β-(1,3)-glucan-synthase enzyme, causing direct damage to the fungal cell wall. Echinocandins-resistance episodes have been attributed to FKS1 or FKS2 genes mutations. Specifically, substitutions alter the already cited genes, producing the target enzyme alteration and the ineffectiveness of the echinocandins [9,10]. Echinocandin-resistance isolates have been rarely reported across European countries. However, a pan-echinocandin-resistant *C. albicans* case report was recently documented from a bloodstream infection in Southern Italy [11,12].

*C. albicans* has been extensively investigated to clarify its drug tolerance mechanisms. First, we can define as tolerance the fungal ability to surviving in drug concentrations above the minimum inhibitory concentration (MIC). The antifungal tolerance may lead to resistance episodes, explaining discrepancies between in vitro susceptibility data and in vivo therapeutical outcomes. Recent published data document heat shock proteins and calcineurin stress-response molecular pathways as possible contributions to the antifungal tolerance. The overexpression or the activation of similar pathways lead to fungal cells’ survival in presence of the antifungal molecules [7].

### 1.2. Antifungal Resistance in Candida glabrata

*C. glabrata* (whose taxonomical current name is *Nakaseomyces glabrata*) expresses an intrinsic low susceptibility to azoles, especially regarding fluconazole, which is the main therapeutical choice in the case of prophylaxis need. The resistance is related to ERG11 gene mutations. Fluconazole-resistant *C. glabrata* emerged across South America, Europe, and Africa. Itraconazole resistance often accompanies these episodes. Some American countries documented increased MIC values with an azole dose-dependent susceptibility for the same species. A similar attitude was reported for voriconazole within Europe and North America. South America revealed fluconazole and miconazole-resistant isolates, while resistance episodes also include miconazole, clotrimazole, itraconazole, and ketoconazole across the Asian South-East region. Finally, Australian regions showed an increasing percentage of fluconazole-resistant isolates [12,13].

Published surveys reveal a significant amphotericin B-resistance incidence in the Asian regions, reporting the same mechanism as *C. albicans*, while elevated epidemiological cut-off values emerged in South America. However, these defined outbreaks did not expand to other world countries, confirming the rare *Candida* spp. tendency to amphotericin B resistance [12,13]. Echinocandin resistance due to FKS genes mutations rarely appeared in *C. glabrata*. Remarkably, Switzerland, Italy, and the United Kingdom reported single *C. glabrata* isolation along with an episode of echinocandin resistance [12,13,14,15].

### 1.3. Antifungal Resistance in Candida parapsilosis

*C. parapsilosis* shows relevant biofilm production, especially in patients carrying central venous catheters or receiving parenteral nutrition. *C. parapsilosis* biofilm reveals high variability, including high carbohydrates and low protein rates within the biofilm extracellular matrix. The biofilm production is one of the most diffused antifungal resistance causes in *C. parapsilosis* isolates, which exhibit decreased antimicrobial susceptibility [16]. Similarly to other *Candida* species, *C. parapsilosis* exhibits different mechanisms related to azole resistance. These mechanisms include the upregulation of the MDR1 efflux pump and alterations in the ergosterol biosynthesis genes such as ERG11. The literature data document fluconazole-resistant *C. parapsilosis* isolation within Central America, Brazil, Southern Europe (especially regarding Italy, Spain, and France), and India [16,17]. Interestingly, *C. parapsilosis* strains recently reported a 1.3% global rate for amphotericin B resistance due to sterol composition variations, ergosterol target replacement (mutations on the ERG1, ERG4, ERG6, and ERG11 genes), and reinforced defences against amphotericin B-related oxidative damage [16]. Finally, echinocandins’ extensive usage facilitated resistance episodes due to FKS genes in *C. parapsilosis* isolates, which frequently show elevated MIC values for these antifungal drugs [16].

### 1.4. Antifungal Resistance in Candida tropicalis

The overexpression of ERG11 is frequently related to azole resistance in *C. tropicalis* strains, allowing a significant increase in 14-lanosterol-demethylases and the survival of the fungal cells. The resistance mechanisms may involve the overexpression of the UPC2 gene, which codifies for a transcription factor related to ergosterol biosynthesis [18]. North America, Latin America, and several European countries documented a moderate resistance rate (<5%) according to recent global surveillance programmes [19]. *C. tropicalis* amphotericin B resistance is very uncommon and related to the already mentioned mechanisms. Experimental data reported amphotericin B-resistant episodes only within North America [20]. Finally, the echinocandin-resistance rate remains <1% in *C. tropicalis*, especially regarding immunocompromised patients within healthcare settings in the USA, India, and Taiwan [21].

### 1.5. Antifungal Resistance in Candida krusei

*C. krusei* (whose taxonomical current name is *Pichia kudriavzevii*) harbours an intrinsic fluconazole resistance, accounting for more than 70% of resistant strains across Europe, North America, Latin America, Asia, and Africa. Voriconazole resistance is less common for this species, reporting approximately 4–14% of resistant isolates within the same geographical areas. Novel triazoles such as posaconazole and isavuconazole still express a relevant antifungal activity against *C. krusei* [22]. Reduced caspofungin susceptibility cases have been reported in North America [23]. Remarkably, there have been records of amphotericin B-resistant *C. krusei* isolates in Asian countries [22,23,24]. Azole, echinocandins, and amphotericin B resistance depends on the already cited mechanisms [22,23,24].

### 1.6. Antifungal Resistance in Uncommon Candida species

According to epidemiological data, several atypical yeast species emerged during the last decade, reporting severe fungal infections among critically ill patients. Clinical practice has to confront insufficient evidence about uncommon species’ antimicrobial susceptibility. *Candida kefyr*, *Candida lusitaniae*, *Candida guilliermondii*, *Candida nivariensis*, and *Candida famata* are the most isolated *Candida* spp. rare species [25].

*C. kefyr* (current taxonomical name *Kluyveromyces marxianus*) has recently been reported in systemic infection episodes, accounting for some multi-drug-resistant strains. Specifically, the literature data reported rare high fluconazole, voriconazole, posaconazole, amphotericin B, and echinocandins MIC values. These uncommon resistant isolates mainly appeared in Kuwait and Turkey [26,27]. The identified resistance mechanisms were the same as for other *Candida* species, along with a relevant biofilm formation tendency.

*C. lusitaniae* (current taxonomical name *Clavispora lusitaniae*) expressed a moderate rate of amphotericin B resistance due to the above-mentioned mechanisms. These episodes were reported in experimental data from Central Europe (France) [25,28]. *C. famata* (whose current taxonomical name is *Debaryomyces hansenii*) rarely causes systemic infections, occasionally demonstrating a reduced susceptibility to azoles and echinocandins in North America [25,29]. *C. guilliermondii* (current taxonomical name *Meyerozyma guilliermondii*) occasionally revealed high antifungal resistance to fluconazole and echinocandins [25,30]. *C. nivariensis* (current taxonomical name *Nakaseomyces nivariensis*) integrates the *C. glabrata* complex as a cryptic fungal species, reporting high virulence and resistance rates. For instance, ERG11 mutations demonstrate a higher incidence than *C. glabrata* [25]. Furthermore, previously published articles documented a *C. nivariensis* therapeutical failure after fluconazole regimens in a Spanish healthcare setting [31].

### 1.7. Antifungal Resistance in Candida auris

*Candida auris* represents a global healthcare challenge due to its extensive antifungal drug resistance. The most intricate feature of this species is the common coexistence of azoles-, echinocandins-, and amphotericin B-resistance mechanisms [32]. Although the specific resistance mechanisms have not been completely clarified, *C. auris* combats the antifungal drugs through the already cited alterations [32]. *C. auris* revealed a significant capability to survive under hard environmental conditions, demonstrating persistence on surfaces and in healthcare settings. These characteristics emphasize the increasing concern about the potential for *C. auris* invasive infections among critical patients [33]. As regards geographical distribution, North America, Brazil, most of the European countries, some Asian regions, and Australia reported pan-drug *C. auris* infection episodes [5,32].

Figure 1, Figure 2 and Figure 3 illustrates the epidemiological incidence of antifungal-resistant *Candida* isolates across the world, the European countries, and Italy. Furthermore, Figure 4 shows the most important antifungal-resistance mechanisms reported in *Candida* spp.

## 2. Results

The evaluation reported a total number of 172 clinical isolates. Globally, the study identified *C. albicans* (82), *C. parapsilosis* (44), *C. glabrata* (18), *C. tropicalis* (16), *C. krusei* (7), *C. lusitaniae* (2), *C. guilliermondii* (1), *C. famata* (1), and *Candida nivariensis* (1). The *Candida* species incidence varied slightly, depending on the analysed period. Specifically, *C. albicans* reached the same incidence rate as non-albicans species in 2020, showing its supremacy in 2021. The non-albicans isolation rate fluctuated during the following years, reporting percentages higher than the *C. albicans* numbers during 2022 and 2024. Otherwise, 2023 demonstrated a relevant (more than 50%) *C. albicans* percentage. Table 1 summarizes all these general data about *C. albicans* and non-*albicans* species.

As regards the non-*albicans* species, *C. parapsilosis* was the most reported species during the analysed period (from 19.5% in 2020 to 36.1% in 2022). On the one hand, 2020 reported a similar *C. glabrata* incidence; on the other hand, *C. parapsilosis* significantly overcame other species during the following years. Notably, *C. famata* appeared in 2021, while *C. guilliermondii* only emerged in 2022.

In addition, *C. lusitaniae* emerged between 2023 and 2024, whereas *C. nivariensis* emerged in 2024. However, these uncommon species were rarely isolated.

Table 2 summarizes *Candida* spp. distribution within the different hospital units during the study period. Remarkably, internal medicine had the most candidaemia episodes. The intensive care (ICU), haematology, and surgery units documented medium candidaemia rates. Finally, the pneumology ward and neonatal intensive care unit (NICU) reported the lowest candidaemia episodes percentages. The uncommon *Candida* species (*C. lusitaniae*, *C. guilliermondii*, *C. famata*, and *C. nivariensis*) reported statistical significance in the hospital units. These species mainly emerged within the pneumology and the internal medicine wards.

Table 3 illustrates details on the *Candida* spp. isolates and their resistance percentages for the analysed antifungal drugs. The authors included amphotericin B, fluconazole, voriconazole, and echinocandins, which represent the most relevant therapeutical choices in the case of invasive candidiasis. The antifungal susceptibility patterns documented several strains with a multi-drug-resistance profile. As regards the *C. albicans* isolates, several strains revealed resistance MIC values for different antifungal drugs.

One strain (1.2%) reported pan-echinocandins resistance, and one strain (1.2%) showed micafungin and anidulafungin resistance. 

Additionally, two strains revealed voriconazole and fluconazole resistance. Among *C. parapsilosis* isolates, one strain showed both fluconazole and voriconazole resistance. One *C. glabrata* isolate showed caspofungin and fluconazole resistance, whereas a single *C. tropicalis* strain revealed fluconazole and voriconazole resistance. One *C. krusei* isolate documented both caspofungin and fluconazole resistance. As regards the uncommon species, *C. nivariensis* and *C. guilliermondii* revealed high fluconazole MIC values. Amphotericin B resistance or non-wild-type isolation did not appear during the evaluation of all the isolated *Candida* species.

## 3. Discussion

Invasive candidiasis is one of the most severe fungal diseases in worldwide hospital settings. Despite several diagnostic improvements, *Candida* spp. bloodstream isolation remains difficult in most cases due to a low sensitivity rate and prolonged turn-around time. Moreover, the lack of global epidemiological data results in significant underestimation of candidaemia cases [34]. Particularly, intensive care patients have an enormous variability in their immunological status, complicating risk-factor identifications. This variability often leads to clinical and diagnostic delays for systemic fungal infections. Published data have demonstrated that infection and mortality rates are frequently underestimated [35].

Based on the relevant candidaemia rate, previously published data analysed the *Candida* spp. distribution within different hospital settings. Xiao et al. reported *C. albicans* as the most isolated *Candida* species in candidaemia cases. *C. parapsilosis*, *C. glabrata*, *C. krusei*, and *C. tropicalis* followed this rate in the same retrospective study, which included different hospital settings in China [36]. Interestingly, our evaluation revealed the same species distribution, which was already documented within larger European evaluations [5]. Our study rarely showed uncommon species such as *C. guilliermondii*, *C. famata*, *C. lusitaniae*, and *C. nivariensis*. These rare isolations complement the literature data [37,38,39,40] and suggest a future focus on uncommon species’ characteristics. Although it has a presence within several European countries, *C. auris* did not emerge across our geographical area. Current data do not exclude possible future appearances due to the extraordinary environmental persistence and diffusion of this species.

Antifungal resistance complicates candidaemia management. Our results confirm the overall presence of a moderate antifungal-resistance percentage. However, some isolates documented challenging resistance mechanisms, such as echinocandin resistance in *C. albicans* and fluconazole resistance in *C. parapsilosis*. Furthermore, azole-resistant *C. albicans* emerged. Previous data rarely reported isolates with the same characteristics, confirming the interest in the collected resistant strains [9,10,11,12]. Despite the typical resistance MIC values, these episodes underline the difficulties in detecting the specific molecular resistance mechanisms. Rare gene mutations may be recognized through advanced generation methodologies, suggesting sequencing analysis’ integration as a final step in the microbiological workflow.

Noticeably, several species revealed multi-drug-resistance episodes. For instance, we analysed *C. glabrata* and *C. krusei* simultaneously reporting caspofungin and fluconazole resistance. These isolates significantly complicate patients’ therapeutical plans, excluding fundamental alternatives in the case of *Candida* spp. dissemination. The literature data emphasized the extreme rarity of these resistance mechanisms’ co-presence and the consequent difficulties in clinical management [22,41]. Similar episodes highlight the significant clinical impact of multi-drug antifungal resistance due to the limited antifungal molecules availability.

The presented data confirmed high fluconazole MIC values for *C. nivariensis* and *C. guilliermondii*, highlighting previously published data [42,43]. The gathered results never described amphotericin B resistance. Previous literature data documented 2–3% of amphotericin B resistance in bloodstream *C. parapsilosis* and *C. krusei* isolates [44]. Additionally, Ahmady et al. stated that *C. lusitaniae* and *C. albicans* may acquire amphotericin B resistance [44]. Despite these rare worldwide isolations, our surveillance did not highlight any similar case. The reported antifungal-resistance cases emphasize the importance of always furnishing precise species identification, especially in the case of severe-infection isolation. Unfortunately, fungal species identification may become challenging in some diagnostic settings. A manuscript from A. Lau documents how essential is to apply advanced identification technologies to avoid misidentification phenomena [45]. Similar data indicates the importance of updating the diagnostic workflow for severe fungal infections. 

This manuscript aims to illustrate antifungal susceptibility data to enrich the epidemiological and literature database about *Candida* spp. resistance patterns. This aim was satisfied through the application of coherent antifungal susceptibility testing in all the systemic infection *Candida* spp. isolations. Unfortunately, Italy is one of the few European countries able to provide antifungal susceptibility data, along with Spain, the United Kingdom, and Norway.

A recent survey reported that most European countries are not currently enabled to execute similar investigations, disallowing the transfer of resistance data to surveillance programmes [46]. This information may suggest the importance of diffuse monitoring of antifungal susceptibility testing methods and practice among European laboratory settings. Surveillance collections often gather resistance data on the first *Candida* spp. blood isolate, avoiding eventual repetitions and bias. However, some isolates develop resistance mechanisms after antifungal drug exposure; thus, secondary collection of the same candidaemia case may be useful to document changes in the susceptibility profile [47,48].

Our study showed possible variability in *Candida* spp. distribution and antifungal susceptibility patterns. The results illustrated how rare antifungal-resistance mechanisms may appear in common *Candida* species, as well as how uncommon *Candida* species may emerge in systemic infection episodes. Complicatedly, the data confirmed the fundamental role of a complete diagnostic workflow, integrating species identification through advanced technologies and MIC values definition through precise methods. Additionally, the possibility of confirming MIC discrepancies through standardized broth microdilution appeared essential due to some discrepancies in commercial methodologies documented by previous published experiences [49,50]. In conclusion, a periodical enrichment of the local epidemiological data may be crucial to prepare different hospital settings for infection control strategies and multi-drug-resistance diffusion in severe fungal infection episodes.

## 4. Materials and Methods

The present manuscript describes a five-year evaluation (2020–2024) on bloodstream infections by *Candida* spp. isolates. The study documents the antifungal susceptibility profiles of *Candida* spp. strains of recovered patients from the University Hospital Policlinico of Catania. Specifically, emergency room, intensive care, internal medicine, infectious diseases, cardiology, transplants, urology, pneumology, and surgery units reported systemic *Candida* spp. isolation. All the isolates were identified using a MALDI Biotyper^®^ Sirius System (Bruker, Billerica, MA, USA). According to current guidelines [51], antifungal susceptibility testing was performed for each bloodstream isolate through the Sensititre™ YeastOne™ YO9 AST Plate (Thermo Fisher Scientific, Waltham, MA, USA). This panel includes echinocandins (micafungin, caspofungin, anidulafungin), azoles (fluconazole, voriconazole, itraconazole, posaconazole), 5-fluocytosine, and amphotericin. The susceptibility profiles were evaluated following the manufacturer’s instructions [52]. Furthermore, the MIC values interpretation was based on the current Clinical and Laboratory Standards Institute guidelines (M27M44S-Ed3 and M57S-Ed4 documents) [51,52]. All the resistance MIC values were confirmed through a CLSI-standardized broth microdilution method [53]. All the above-mentioned procedures did not involve direct interventions on human beings and only regarded clinical isolates. 

The authors reported a statistical evaluation of *Candida* spp. distribution depending on the analysed hospital units. They applied the MedCalc Statistical Software version 17.9.2 (MedCalc Software bvba, Ostend, Belgium; http://www.medcalc.org; 2017, accessed on 30 June 2024), reporting the corresponding *p* values. The χ2 and Fisher’s exact test established the categorical variables as percentages. Furthermore, the study included an analysis of the most isolated *Candida* spp. for each study year. Finally, details on the antifungal susceptibility profiles were summarized in tables. The categories “susceptible” (S), “intermediate” (I), “susceptible dose-dependent” (SDD), and “resistant” (R) included the isolates according to the CLSI guidelines. Epidemiological cut-off (E-COFF) allowed *Candida* spp. classification as wild-type (WT, with a MIC value equal or lower than the E-COFF) or non-wild-type (non-WT, with a MIC value higher than the E-COFF) in the absence of a clinical breakpoint. The wild-type isolates were considered presumptively susceptible, while the non-wild-type strains were related to a hypothetical resistance.

## Figures and Tables

**Figure 1 antibiotics-13-00914-f001:**
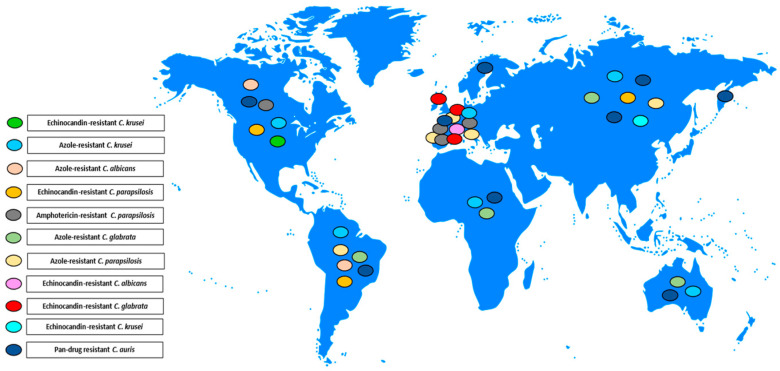
Antifungal-resistant *Candida* spp. isolates distribution across the world [5,6,7,8,9,10,11,12,13,14,15,16,17,18,19,20,21,22,23,24,25,29,32]. Coloured marks signal these isolates’ presence within the countries according to the reported colours legend.

**Figure 2 antibiotics-13-00914-f002:**
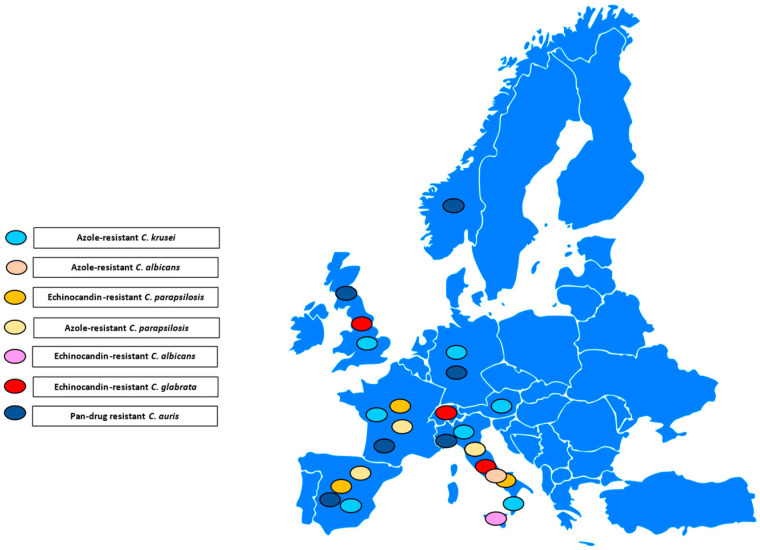
Antifungal-resistant *Candida* spp. isolates distribution across the European countries [5,6,7,8,9,10,11,12,13,14,15,16,17,18,19,20,21,22,23,24,25,29,32]. Coloured marks signal these isolates’ presence within the countries according to the reported colours legend.

**Figure 3 antibiotics-13-00914-f003:**
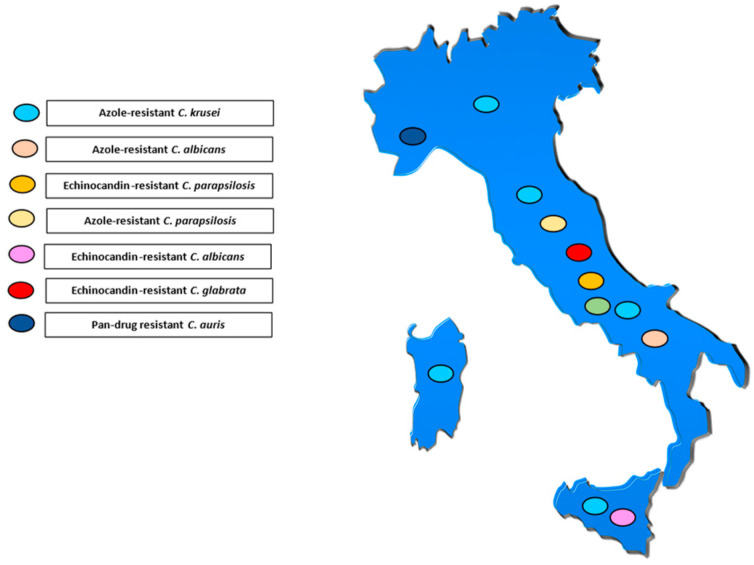
Antifungal-resistant *Candida* spp. isolates distribution across Italy [5,6,7,8,9,10,11,12,13,14,15,16,17,18,19,20,21,22,23,24,25,29,32]. Coloured marks signal these isolates’ presence within the country according to the reported colours legend.

**Figure 4 antibiotics-13-00914-f004:**
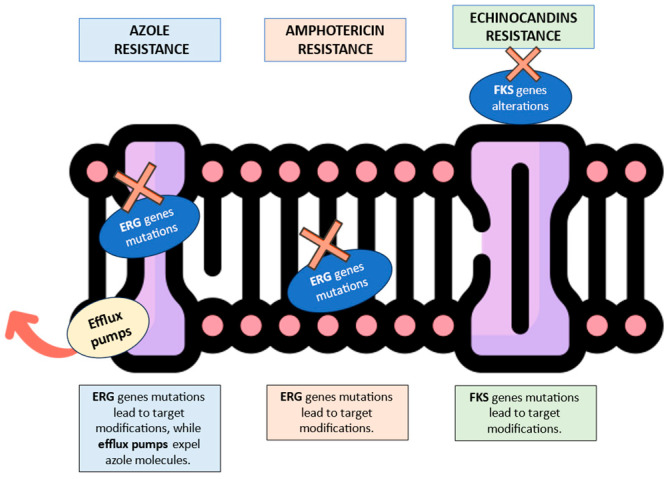
Graphical summary of the most important antifungal-resistance mechanisms reported in *Candida* spp. The red “X” symbols indicate the resistance mechanism in its specific cell location. The red arrow illustrates the efflux pump’s capability to expel antifungal molecules from the fungal cell (https://www.flaticon.com/search?wordcell%20membrane, accessed on 27 August 2024).

**Table 1 antibiotics-13-00914-t001:** Distribution of *Candida* species by year.

*Candida* Species	Total (%)	2020 (%)	2021 (%)	2022 (%)	2023 (%)	2024 (%)	*p*
*Candida albicans*	82 (47.7)	20 (48.8)	22 (55.0)	16 (44.4)	14 (58.3)	10 (32.2)	0.279
*Candida parapsilosis*	44 (25.6)	8 (19.5)	10 (25.0)	13 (36.1)	4 (16.7)	9 (29.0)	0.393
*Candida glabrata*	18 (10.5)	6 (14.6)	4 (10.0)	3 (8.3)	1 (4.2)	4 (12.9)	0.706
*Candida krusei*	7 (4.1)	4 (9.7)	0	0	2 (8.3)	1 (3.2)	0.099
*Candida tropicalis*	16 (9.3)	3 (7.3%)	3 (7.5)	3 (8.3)	2 (8.3)	5 (16.1)	0.712
*Candida guilliermondii*	1 (0.6)	0	0	1 (2.7)	0	0	0.433
*Candida famata*	1 (0.6)	0	1 (2.5)	0	0	0	0.505
*Candida lusitaniae*	2 (1.2)	0	0	0	1 (4.2)	1 (3.2)	0.353
*Candida nivariensis*	1 (0.6)	0	0	0	0	1 (3.2)	0.333
Total	172 (100)	41 (23.8)	40 (23.2)	36 (20.9)	24 (13.9)	31 (18.0)	

**Table 2 antibiotics-13-00914-t002:** Distribution of *Candida* species according to different hospital units.

*Candida* Species	Total (%)	ICU	NICU	Hematology ^a^	Surgery	Pneumology	Internal Medicine	*p* Value
*Candida albicans*	82 (47.7)	20 (39.2)	0	4 (30.8)	16 (53.3)	5 (50.0)	37 (49.3)	0.334
*Candida parapsilosis*	44 (25.6)	10 (19.6)	2 (66.7)	4 (30.8)	7 (23.3)	0	21 (28.0)	0.177
*Candida glabrata*	18 (10.5)	6 (11.8)	0	1 (7.7)	5 (16.7)	2 (20.0)	4 (5.3)	0.410
*Candida krusei*	7 (4.1)	0	1 (33.3)	2 (15.4)	0	0	4 (5.3)	0.107
*Candida tropicalis*	16 (9.3)	4 (7.8)	0	2 (15.4)	2 (6.7)	1 (10.0)	7 (9.3)	0.938
Others ^b^	5 (2.9)	1 (1.9)	0	0	0	2 (20.0)	2 (2.7)	0.02
Total	172 (100)	51 (29.6)	3 (1.7)	13 (7.5)	30 (17.4)	10 (5.8)	75 (43.6)	

^a^ Adult and paediatric haematology and transplantation. ^b^ Other *Candida* species are *C. lusitaniae*, *C. guilliermondii*, *C. famata*, *C. nivariensis*.

**Table 3 antibiotics-13-00914-t003:** Antifungal susceptibilities to antifungal agents for various *Candida* species.

*Candida* Species and Antifungal Agent	Minimum Inhibitory Concentration			In Vitro Susceptibility; No. (%)			
	Range (mg/L)	50	90	S	I	SDD	R
*Candida albicans* (82)							
Amphotericin B	<0.12–1	0.25	0.5	82 (100) ^a^	0	0	0
Fluconazole	<0.12–256	0.25	0.5	80 (97.6)	-	0	2 (2.5)
Anidulafungin	<0.008–1	0.015	0.12	80 (97.6)	0	-	2 (2.5)
Micafungin	<0.008–4	0.015	0.03	80 (97.6)	0	-	2 (2.5)
Caspofungin	0.008–>8	0.03	0.12	81 (98.8)	0	-	1 (1.2)
Voriconazole	<0.008–>8	0.008	0.015	80 (97.6)	0	-	2 (2.5)
*Candida parapsilosis* (44)							
Amphotericin B	<0.12–1	0.25	0.5	44 (100) ^a^	0	0	0
Fluconazole	0.06–>128	0.5	8	36 (81.8)	-	2 (4.5)	6 (13.6)
Anidulafungin	0.5–4	0.5	2	43 (97.7)	1 (2.3)	-	0
Micafungin	0.015–2	1	2	44 (100)	0	-	0
Caspofungin	0.12–2	0.5	1	44 (100)	0	-	0
Voriconazole	<0.008–1	0.015	0.25	39 (88.6)	4 (9.1)	-	1 (2.3)
*Candida glabrata* (18)							
Amphotericin B	<0.12–1	0.5	1	18 (100) ^a^	0	0	0
Fluconazole	4–>256	16	128	-	-	15 (83.3)	3 (16.6)
Anidulafungin	<0.015–0.12	0.03	0.06	18 (100)	0	-	0
Micafungin	0.015–0.03	0.015	0.03	18 (100)	0	-	0
Caspofungin	0.03–0.5	0.06	0.12	16 (88.8)	1 (5.5)	-	1 (5.5)
Voriconazole	0.25–8	0.5	2	3 (16.6%) ^a^	0	0	15 (83.3%) ^a^
*Candida krusei* (7)							
Amphotericin B	<0.12–1	*	*	7 (100) ^a^	0	0	0
Fluconazole	64–128	*	*	0	-	0	7 (100)
Anidulafungin	0.03–0.12	*	*	7 (100)	0	-	0
Micafungin	0.12–0.25	*	*	7 (100)	0	-	0
Caspofungin	0.06–1	*	*	4 (57.1)	2 (28.6)	-	1 (14.3)
Voriconazole	0.25–1	*	*	7 (100)	0	-	0
*Candida tropicalis* (16)							
Amphotericin B	<0.12–1	0.5	1	16 (100) ^a^	0	0	0
Fluconazole	0.5–8	2	4	13 (81.2)	-	2 (12.5)	1 (6.2)
Anidulafungin	<0.015–0.12	0.06	0.12	16 (100)	0	-	0
Micafungin	<0.008–0.06	0.03	0.06	16 (100)	0	-	0
Caspofungin	0.008–0.5	0.06	0.12	15 (93.7)	1 (6.2)	-	0
Voriconazole	0.015–1	0.125	0.25	8 (50)	7 (43.7)	-	1 (6.2)
*Candida guilliermondii* (1)							
Amphotericin B	1	*	*	1 (100) ^a^	0	0	0
Fluconazole	64	*	*	1 (100)	0	0	0
Anidulafungin	0.12	*	*	1 (100)	0	-	0
Micafungin	0.25	*	*	1 (100)	0	-	0
Caspofungin	0.12	*	*	1 (100)	0	-	0
Voriconazole	1	*	*	IE	IE	IE	IE
*Candida famata* (1)							
Amphotericin B	<0.12	*	*	IE	IE	IE	IE
Fluconazole	4	*	*	IE	IE	IE	IE
Anidulafungin	0.5	*	*	IE	IE	IE	IE
Micafungin	0.25	*	*	IE	IE	IE	IE
Caspofungin	0.06	*	*	IE	IE	IE	IE
Voriconazole	0.12	*	*	IE	IE	IE	IE
*Candida lusitaniae* (2)							
Amphotericin B	<0.12–0.25	*	*	2 (100) ^a^	0	0	0
Fluconazole	1	*	*	2 (100)	0	0	0
Anidulafungin	0.03–0.12	*	*	2 (100)	0	0	0
Micafungin	0.015–0.125	*	*	2 (100)	0	0	0
Caspofungin	0.06–0.5	*	*	2 (100)	0	0	0
Voriconazole	0.015	*	*	IE	IE	IE	IE
*Candida nivariensis* (1)							
Amphotericin B	0.5	*	*	IE	IE	IE	IE
Fluconazole	8	*	*	IE	IE	IE	IE
Anidulafungin	0.03	*	*	IE	IE	IE	IE
Micafungin	0.03	*	*	IE	IE	IE	IE
Caspofungin	0.06	*	*	IE	IE	IE	IE
Voriconazole	0.25	*	*	IE	IE	IE	IE

Abbreviations: S, susceptible; I, intermediate; SDD, dose-dependent susceptible; R, resistant; IE, insufficient evidence. * MIC50 and MIC90 values were not performed because the number is smaller than 10. ^a^ A presumptive susceptibility or resistance was reported according to the CLSI amphotericin B E-COFF values.

## Data Availability

All the gathered data were included in the manuscript.

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
