# Peer review of "Antifungal Susceptibility Data and Epidemiological Distribution of Candida spp.: An In Vitro Five-Year Evaluation at University Hospital Policlinico of Catania and a Comprehensive Literature Review"

_antibiotics, 2024, doi:10.3390/antibiotics13100914_

Round 1

Reviewer 1 Report

Comments and Suggestions for Authors

Overall, the manuscript offers a valuable analysis of antifungal susceptibility and the epidemiological distribution of Candida species over five years at the University Hospital Policlinico of Catania. The data presented provides meaningful insights for readers, particularly in understanding species distribution (abundance) and susceptibility patterns to various antifungal agents.

However, some revisions are necessary to strengthen the manuscript further.

in general: The discussion section is currently too brief and lacks depth. It should be expanded to provide a more comprehensive analysis of the findings. For example: compare these results to more similar studies highlighting the emergence of the new species (like C. auris) in other studies and the potential cause for that.

Details:

Title: the hospital name (University Hospital Policlinico of Catania) must be added to the title.

Figures 1, 2, and 3: The references for the data presented in these figures are missing. Please include the appropriate references. Additionally, Reference 33 does not appear to be relevant to the distribution in the world, Europe, and Italy.

I don’t see any detection of the emerging species C. auris. Is this normal or the identification tools cannot detect it?

Line 26: use “fungal virulence and/or patients’ underlying” instead of “fungal virulence or patients’ underlying”.

Line 44: use “specific options” instead of “targeted options”.

Lines 48-50: This is just repetition of what is exactly mentioned in the abstract. Please remove or rephrase.

Line 55: use “Lanosterol 14-α-Demethylase” instead of “ lanosterol-demethylase”.

Line 56: use “ERG11” instead of “lanosterol” since ERG11 is the target not lanosterol.

Lines 56-58: this information is incorrect it is talking about azole resistance in C. auris as rare. However, this should be echinocandin resistance and the references are for echinocandins so please correct.

Line 63-58: This is a sudden shift between 5FC and AmB resistance. Please separate them and clarify the causes and consequences for each.

Line 72: use “fungal cell wall” instead of “fungal cell membrane”

Line 78: reference no 12 is not relevant to the sentence.

Lines 79-87: This paragraph is not well written please remove or enhance it.

Line 96: use “accompanies” instead of “accompain”.

Line 175: Remove “Sensu strictu”.

Table 1: Candida tropicals 3 in 2020. Please provide the % as others.

Line 293: Please mention that this study is performed in different hospital settings in China. If you can add data from Europe and compare, it would be great. 

Comments on the Quality of English Language

No comment.

Author Response

Comment: In general: The discussion section is currently too brief and lacks depth. It should be expanded to provide a more comprehensive analysis of the findings. For example: compare these results to more similar studies highlighting the emergence of the new species (like C. auris) in other studies and the potential cause for that.

            Answer: Thank you for these comments. The discussion section has been improved, also considering few words about C. auris and its current absence within our geographical area.

Comment: Title: the hospital name (University Hospital Policlinico of Catania) must be added to the title.

            Answer: The hospital name has been added to the title.

Comment: Figures 1, 2, and 3: The references for the data presented in these figures are missing. Please include the appropriate references. Additionally, Reference 33 does not appear to be relevant to the distribution in the world, Europe, and Italy.

            Answer: The references have been added to the figures. The addition of reference 33 within the geographical distribution paragraph was a typo. The correct reference (32) has been added.

Comment: I don’t see any detection of the emerging species C. auris. Is this normal or the identification tools cannot detect it?

            Answer: C. auris identification is possible through mass spectrometry technologies, which have been integrated in our diagnostic routine several years ago. However, this species has not emerged yet within our geographical area.

Comment: Line 26: use “fungal virulence and/or patients’ underlying” instead of “fungal virulence or patients’ underlying”.

            Answer: Thank you for the suggestion. It has been corrected.

Comment: Line 44: use “specific options” instead of “targeted options”.

            Answer: It has been corrected.

Comment: Lines 48-50: This is just repetition of what is exactly mentioned in the abstract. Please remove or rephrase.

            Answer: Sorry for the repetition. Lines 48-50 have been rephrased.

Comment: Line 55: use “Lanosterol 14-α-Demethylase” instead of “lanosterol-demethylase”.

            Answer: It has been corrected.

Comment: Line 56: use “ERG11” instead of “lanosterol” since ERG11 is the target not lanosterol.

            Answer: It has been corrected.

Comment: Lines 56-58: this information is incorrect it is talking about azole resistance in C. auris as rare. However, this should be echinocandin resistance and the references are for echinocandins so please correct.

            Answer: Sorry for the error. Lines 56-58 regard C. albicans azole-resistance, so the correct references are 7,8. They have been corrected.

Comment: Line 63-58: This is a sudden shift between 5FC and AmB resistance. Please separate them and clarify the causes and consequences for each.

            Answer: These sections have been slightly separated. Some words have been added to better clarify the impact of those antifungal resistance.

Comment: Line 72: use “fungal cell wall” instead of “fungal cell membrane”

            Answer: It has been corrected.

Comment: Line 78: reference no 12 is not relevant to the sentence.

            Answer: We decided to integrate references 11 and 12 at the end of the same sentence, as literature data confirmations of the rare echinocandin-resistance phenomenon in C. albicans.

Comment: Lines 79-87: This paragraph is not well written please remove or enhance it.

            Answer: The paragraph has been simplified. Thank you for the suggestion.

Comment: Line 96: use “accompanies” instead of “accompain”.

            Answer: It has been corrected.

Comment: Line 175: Remove “Sensu strictu”.

            Answer: It has been removed.

Comment: Table 1: Candida tropicalis 3 in 2020. Please provide the % as others.

            Answer: It has been provided.

Comment: Line 293: Please mention that this study is performed in different hospital settings in China. If you can add data from Europe and compare, it would be great.

            Answer: The paragraph has been modified according to the suggestions.

Reviewer 2 Report

Comments and Suggestions for Authors

Dear authors, congratulations on the excellent manuscript. It addresses a very important topic and serves as a basic reading for all researchers who work with Candida species. Below I will leave some revisions so that the manuscript can be improved.

1 - I suggest that you make didactic illustrations showing the mechanisms of Candida resistance.

2 - Carefully check all scientific names, which should be in italics. See the example in line 230.

3 - Some Candida species have changed gender, so that they are no longer part of the group. Carefully review your manuscript; one example is Candida krusei, now classified as Pichia kudriavzevii.

4 - The text is repetitive in several sections, especially when discussing the mechanisms of antifungal resistance among different Candida species. For example, the resistance mechanisms related to the ERG11 gene and the use of efflux pumps are repeatedly mentioned without adding significant new information. This makes the reading tiresome and dilutes the relevance of the main points. 5- The text makes broad statements, such as "Resistance episodes frequently involve 5-fluocytosine" or "Epidemiological studies have mainly reported...". These generalizations lack specificity and support from concrete data, which weakens the credibility of the statements.

6- Although the text describes antifungal resistance, it fails to critically address the clinical implications or propose innovative solutions. There is a lack of in-depth analysis of how these findings could impact clinical practice or future research, which limits the contribution of the article to the field.

Author Response

1 – Comment: I suggest that you make didactic illustrations showing the mechanisms of Candida resistance.

            Answer: Thank you for the suggestion. Figure 4 has been added to summarize the main antifungal resistance mechanisms.

2 – Comment: Carefully check all scientific names, which should be in italics. See the example in line 230.

            Answer: Sorry for the typos. They have been corrected.

3 – Comment: Some Candida species have changed gender, so that they are no longer part of the group. Carefully review your manuscript; one example is Candida krusei, now classified as Pichia kudriavzevii.

            Answer: thank you for the essential suggestion. We decided to add the current taxonomical name in brackets for all the cited Candida species. We maintained the other name (such as Candida krusei along with Pichia kudriavzevii) because the taxonomy databases consider the two expressions as synonyms.

4 – Comment: The text is repetitive in several sections, especially when discussing the mechanisms of antifungal resistance among different Candida species. For example, the resistance mechanisms related to the ERG11 gene and the use of efflux pumps are repeatedly mentioned without adding significant new information. This makes the reading tiresome and dilutes the relevance of the main points. 5- The text makes broad statements, such as "Resistance episodes frequently involve 5-fluocytosine" or "Epidemiological studies have mainly reported...". These generalizations lack specificity and support from concrete data, which weakens the credibility of the statements.

            Answer: Thanks for the suggestions. Some repetitions have been removed. Broad statements have been rephrased.

6- Comment: Although the text describes antifungal resistance, it fails to critically address the clinical implications or propose innovative solutions. There is a lack of in-depth analysis of how these findings could impact clinical practice or future research, which limits the contribution of the article to the field.

            Answer: Thank you for the suggestions. We revised the discussion section, adding some sentences to highlight possible implications and improvements. 

Reviewer 3 Report

Comments and Suggestions for Authors

1. Authors describe different levels of susceptibility of several Candida species, one by one, to antifungals mainly azoles and echinocandins, depending on the world region and some nosocomial areas. Their work is an evaluation of what has been published in this subject without contributing with new ideas  to explain the basis of this geographic variability of human susceptibility to candidiasis and antifungal resistance. 

2. Only four out of the 53 enlisted references include one or all three authors in different positions. This means a short experience in the subject.

3. Several organism names in the reference list do not appear in italics. Please edit. 

Comments on the Quality of English Language

English is fine. 

Author Response

  1. Comment: Authors describe different levels of susceptibility of several Candida species, one by one, to antifungals mainly azoles and echinocandins, depending on the world region and some nosocomial areas. Their work is an evaluation of what has been published in this subject without contributing with new ideas to explain the basis of this geographic variability of human susceptibility to candidiasis and antifungal resistance.

            Answer: Thank you for the considerations. The main purpose of our study was to share local epidemiological data, highlighting the clinical impact of antifungal resistance and the extreme importance to always provide details about antifungal susceptibility and species identification. Additionally, the literature review completed the study to integrate our local data in a more complex and wider epidemiological context.

  1. Comment: Only four out of the 53 enlisted references include one or all three authors in different positions. This means a short experience in the subject.

            Answer: Thank you for the observation. Actually, all the authors work in the clinical microbiology field since several years. They have been playing different roles into the diagnostic micology section, also including expertise on antifungal resistance and its clinical impact. However, the study aimed to compare local data to literature data about other regions and countries. Consequently, it was essential to insert larger studies, extensive reviews, and influential data from other qualified authors. 

  1. Comment: Several organism names in the reference list do not appear in italics. Please edit.

            Answer: Sorry for the typos. They have been corrected.

Round 2

Reviewer 1 Report

Comments and Suggestions for Authors

No further comments. I recommend accepting the paper.